# Effects of Standardized Patient Simulation and Mobile Applications on Nursing Students’ Clinical Competence, Self-Efficacy, and Cultural Competence: A Quasi-Experimental Study

**DOI:** 10.3390/ijerph21040515

**Published:** 2024-04-22

**Authors:** Duc Nu Minh Ton, Trang Thi Kieu Duong, Hang Thi Tran, Thanh Thi Thanh Nguyen, Hai Ba Mai, Phuong Thi Anh Nguyen, Binh Duy Ho, Trang Thi Thuy Ho

**Affiliations:** 1Faculty of Nursing, Hue University of Medicine and Pharmacy, Hue University, Hue 49000, Vietnamdtktrang@dhktyduocdn.edu.vn (T.T.K.D.); maibahai@hueuni.edu.vn (H.B.M.); ntaphuong@hueuni.edu.vn (P.T.A.N.); 2Faculty of Nursing, Da Nang University of Medical Technology and Pharmacy, Da Nang 550000, Vietnam

**Keywords:** standardized patient simulation, mobile applications, nursing education, cultural competence, nursing clinical performance, self efficacy

## Abstract

Background: Simulation-based education has emerged as an effective approach in nursing education worldwide. We aimed to evaluate the effectiveness of a surgical nursing education program based on a simulation using standardized patients and mobile applications among nursing students. Methods: A mixed-methods design with a quasi-experimental longitudinal approach and focus group interviews was employed. The data were collected from 130 third-year nursing students at three different time points who were equally divided into experimental and control groups. This study measured the level of clinical surgical nursing competence, self-efficacy in clinical performance, cultural competence, and satisfaction with simulation experience. Four focus group interviews were conducted using open-ended questions to explore the participants’ perspectives on the course’s efficacy and satisfaction. Results: There were statistically significant differences in clinical surgical nursing competence (F = 8.68, *p* < 0.001), self-efficacy in clinical performance (F = 13.56, *p* < 0.001), and cultural competence (F = 10.35, *p* < 0.001) across time between the intervention and control groups. Student satisfaction with the simulation-based training was high, particularly regarding debriefing and reflection, with an overall mean satisfaction level of 4.25 (0.40). Students’ perspectives regarding integrated hybrid training are categorized into three themes: educational achievement, dynamic learning experiences, and satisfaction and suggestion. Conclusion: Simulation-based learning provides a dynamic and immersive educational experience that enables undergraduate nursing students to develop and refine essential clinical skills while also fostering confidence and cultural competence.

## 1. Introduction

Clinical and cultural competence are essential in the humanization of nursing care [1,2,3] to achieve improved patient outcomes, well-being, quality of life, and professional performance. Notably, the provision of adequate care and attention to surgical patients during the perioperative phase is one of the fundamental responsibilities of nurses. It involves comprehensive assessment and preoperative emotional and physical preparation, anesthesia safety, surgical positioning, skin preparation, temperature regulation, airway, hemodynamic stability, pain, and symptomatology management in the transoperative and postoperative phases [4]. Thus, nursing care is a holistic endeavor that goes beyond the technical aspects of healthcare. Nurses must serve as scientific and moral agents, integrating their clinical expertise with compassion, empathy, and ethical considerations to promote humanized care [5,6]. It is essential to incorporate a comprehensive curriculum that integrates effective learning experiences in the clinical setting for nursing students to ensure that they are well-prepared for the nursing profession.

Simulation-based nursing education is a pedagogical approach that offers a dynamic and active learning experience by providing a safe learning environment and preparing for challenges facing clinical practice [7,8]. Standardized patient simulation, in particular, plays a crucial role in creating immersive and realistic scenarios in nursing education [8,9]. Integrating standardized patients into simulation-based nursing education significantly enhances the authenticity and consistency of scenarios, allowing the transfer of theoretical knowledge to practical settings [10,11]. This approach facilitates the development of essential skills and competencies while also fostering a deep understanding of the art and essence of nursing by acknowledging the holistic nature of human beings and delivering patient-centered care [11]. Previous studies have demonstrated the effectiveness of simulation-based nursing education in developing a wide range of skills and competencies crucial for nursing practice, including developing practical skills, reacting and thinking in real situations, communication skills, clinical practice competency, empathy, and self-confidence to provide humanized nursing care [10,12].

Mobile apps are increasingly becoming an integral component of nursing education owing to their diverse applications in both educational and clinical contexts [13,14]. They serve as a valuable adjunct to learning modalities, enabling students to perform multiple tasks simultaneously by leveraging technological advancements [15]. A plethora of nursing apps, such as laboratory testing guides, drug interaction guides, nursing care guides, and nursing assessment guides, are widely used to improve patient care [14]. These quick and convenient resources provide nursing students with vital information, and their use is expected to increase as nursing becomes more complex [13,15,16]. Previous studies have demonstrated that the use of mobile apps among nursing students facilitates increased knowledge, skills, and motivation as well as the provision of safe, evidence-based care to patients [15,17]. Within this context, the expansion of technology integration in educational programs is essential to equip students with digital health skills, to be adept at leveraging technology to deliver quality patient care, and to prepare them for the challenges of the future workplace [13,14].

Therefore, this study was designed to identify the effectiveness of a surgical nursing education program based on a simulation using standardized patients and a mobile application that allows nursing students to experience improved preparedness, nursing competency, and cultural competence. The effectiveness of the program was subsequently investigated and evaluated.

## 2. Materials and Methods

### 2.1. Study Design and Participants

A concurrent triangulation mixed-methods design, with a quasi-experimental longitudinal design and focus group from December 2022 to June 2023, was used in this study. The data were collected at baseline before the intervention (T0), immediately after the intervention (T1), and at the four-week follow-up (T2) to evaluate the effects of the intervention (Figure 1).

The current study was conducted at X University in Vietnam. A total of 130 participants were screened for enrolment in the study. The sample size was calculated using G*Power version 3.1.9.4 software, effect size = 0.5, α = 0.25, and power = 0.95, and 130 participants were required for calculation. Third-year nursing students were recruited and allocated to the experimental (n = 65) or control group (n = 65). Of these, 60 participants in the control group completed the full 8 weeks, and 2 were lost to baseline and follow-up in the experimental group (stopped attending sessions or providing data). No incentives to participate in the study were offered to any participants.

The research team conducted focus group interviews after the intervention (T1) with four groups, each consisting of four to six participants. The interviews were held in a seminar room at a university and lasted for two hours per group. Two researchers facilitated each group, while one researcher acted as a note-taker to ensure accurate recording of the discussions. At the end of each interview, the interviewer summarized the conversation and confirmed key information for data accuracy.

### 2.2. Instruments for Training Evaluation

Qualitative and quantitative data analyses were conducted in the evaluation process based on the guidelines of the NLN/Jeffries Nursing Education Simulation Framework.

#### 2.2.1. Quantitative Measures

The authors developed a clinical surgical nursing competence scale to determine the competence of clinical surgical nursing students The tool included 23 items with 4 Likert-type scales ranging from 1 = strongly disagree to 4 = strongly agree. It included 4 subscales: surgical operation nursing care; surgical nursing health education; wound care; and foundational nursing. The scale was developed mainly based on current surgical nursing course outcomes of the university along with existing self-assessment scales of clinical competence among nursing students including Perceived Perioperative Competence Scale-Revised and Nurse Professional Competence Scale [18]. Examples of some states in this tool are as follows: “I have the knowledge and ability to provide care for pre and postoperative patients”, and “I have the knowledge and ability to provide health education on monitoring and preventing surgical site infections”. This scale was then sent to two different experts for revision with CVI values ranging from 0.70 to 1.00. The pilot study was conducted afterward to evaluate before using for the main study. In this study, Cronbach’s alpha of the overall scale was 0.91, indicating high internal consistency.

Self-efficacy in clinical performance scales was measured [19]. It contains one set of 37 items with four subscales: assessment, diagnosis, planning, implementation, and evaluation. The total score ranges from 37 to 185, where a higher score indicates greater self-efficacy in clinical performance. The Vietnamese-translated and -validated version was used in the pilot study, with CVI values ranging from 0.75 to 1.00. The Cronbach’s α of this scale was 0.96 in the original study [19] and 0.97 in this study.

The nurse cultural competence scale was a broad, multidimensional instrument for the detailed and comprehensive measurement of cultural competence [20]. The instrument consists of 20 items with a 5-point Likert scale, where 5 means “I strongly agree” and 1 means “I strongly disagree”. This tool obtained satisfactory psychometric properties and reliability (internal consistency of Cronbach’s alpha 0.95) in our study.

The satisfaction with simulation experience scale was used to assess overall satisfaction with the simulation activities [21]. This tool consists of 18 items with three subscales: debriefing and reflection, clinical reasoning, and clinical learning. The items are rated on a 5-point Likert-type scale ranging from strongly disagree to strongly agree. The Vietnamese-translated and -validated versions were used [22]. The Cronbach’s alpha of the total scale in our study was 0.95.

The instruments were translated by the researcher from English into Vietnamese through the process of translation [23] and revised through comments from two expert panels. Additionally, forward–backwards translation was used for the qualitative data.

#### 2.2.2. Qualitative Measures

Four focus group interviews were sufficient to explore the effects of the course. Open–closed questions were used at the learning level to understand participants’ perceptions, comprising the following: *Please tell me your experience attending the simulation course. How does the course increase your awareness/knowledge/skill/clinical competence, and self-efficacy in clinical performance/cultural competence? How were you able to meet the objectives? What else do you want to know about clinical competence? Do you have any other comments about your experience in this course?*

Focus group questions evaluating satisfaction included the following: Tell me about your simulation course satisfaction (content, material, instructors, facilities). What teaching approaches/materials are the most effective/ineffective? What support might you need to apply what you learned?

### 2.3. The Development of Surgical Nursing Training

#### 2.3.1. Control Group

The control group attended a four-week training program at a clinical hospital and utilized mobile healthcare application apps during practice. Mobile healthcare applications intended for use in comprehensive nursing assessment include collecting objective data, mobile wound care, nursing care plans, and disease management or prevention. These mobile healthcare applications were applied including HSCC [24]; Mdcalc [25]; Nursing Diagnosis and Care Plans App [26]; and imitoMeasure [27]; Mobile healthcare applications have been utilized significantly to enhance nursing assessment by enabling the collection of objective data. The application has also shown promise in the realm of wound care, where they provide an effective means of monitoring and tracking the healing progress of wounds while offering guidance on wound care techniques and practices. Moreover, mobile applications facilitated the development and management of nursing care plans, enabling students to create tailored care plans for individual patients. The participants also used mobile healthcare apps to support the management or prevention of disease by guiding healthy living while also offering disease management tools that enable patients to monitor and track symptoms and manage medications.

#### 2.3.2. Experimental Group

The participants in the experimental group underwent a four-week training program at a clinical hospital. During the first three-week period, the participants also engaged in four simulation training sessions and utilized mobile healthcare applications in both the simulation practice and clinical settings. The participants attended four clinical scenarios, and each clinical scenario session lasted for 120–150 min, depending on the topic. The students were divided into eight groups, each consisting of seven or nine students.

#### 2.3.3. Description of the Intervention

The course was developed following the systematic approach of the ADDIE model from Analyze, to Design, Develop, Implement, and ultimately Evaluate. The results of the analysis stage show that although the current readiness among nursing students is moderate, a proactive approach to implementation and maintenance is crucial [28]. Therefore, nursing educators could ensure that simulation becomes an integral and effective component of nursing education, ultimately enhancing the overall preparedness of future healthcare professionals. As a result, in our institution, simulation was initially introduced only in surgical nursing clinical courses.

This study used the NLN/Jeffries Nursing Education Simulation Framework to guide the process of planning, conducting, and evaluating simulation activities. The framework provides a theoretical direction for aligning simulation activities with learning outcomes, addressing diverse student needs, and ensuring simulation quality and rigor [29]. This framework has been widely adopted in the field of nursing education to improve learning and performance outcomes [30].

The simulation scenario template was developed based on the California Simulation Alliance’s simulation scenario template, the National League for Nursing simulation scenario template, and guidelines for writing a simulation scenario [31,32,33]. The surgical nursing clinical course development team formulated four clinical scenario lesson plans based on two topics: perioperative nursing care for kidney stones and perioperative nursing care for bile duct stones. Clinical nursing and medical experts in each medical field were invited to validate and revise the teaching scenarios. The scenarios were revised again to form the final version.

For each scenario, a 120 min session included prebriefing (10 min), simulation activity (90 min), and debriefing (20 min). The standardized patients were carefully trained to accurately portray the characteristics and symptoms of actual patients, providing students with an immersive and realistic learning experience. Finally, to assess the student’s performance, educators used a combination of the Lasater Clinical Judgment Rubric [34] and reflective writing. By incorporating these assessment approaches, educators could gain a thorough understanding of each student’s strengths and weaknesses, allowing for targeted feedback and improved learning outcomes.

Additionally, patient data were collected via mobile applications for nursing assessment. Nursing care plan apps were supported to manage treatment plans. Other health apps were used to educate patients and promote healthy habits.

#### 2.3.4. Ethical Considerations

This research followed the Declaration of Helsinki, and the Ethics Committee of X University approved the protocol (H2022/113). All participants in this study consented to complete the survey and participate in the research by providing a written agreement after reviewing the recruitment details and research participation instructions. Additionally, this manuscript does not include any animal studies, nor does it contain any potentially identifiable images or data.

### 2.4. Data Collection

The participants for this study were selected and underwent a pretest at the beginning of the course. They completed a posttest after the course and then a follow-up test after 4 weeks. Before agreeing to participate and providing their personal information, all participants were asked to carefully review the purpose and content of the study, confidentiality agreements, and their right to withdraw. Each participant was given a small incentive, which was provided by the survey regulations.

### 2.5. Data Analysis

This study used SPSS 26.0 software (SPSS Inc., Chicago, IL, USA). Descriptive statistics were used to describe the characteristics of the participants. Chi-square test, Fisher’s exact test, *t*-test, and mixed models were applied to compare the data.

To analyze the qualitative data, ATLAS.ti version 8.0 was utilized for content analysis. The qualitative content analysis process involves analyzing data, performing careful follow-up on the entire analysis process and categorization for the organization phase, and explaining the categorization process by using tables and quotations for the reporting phase [35]. Additionally, this study applied the process of ensuring trustworthiness [36].

## 3. Results

### 3.1. Quantitative Results

#### 3.1.1. Homogeneity Test of General Characteristics and Variables at Baseline

A total of 124 third-year nursing students were involved in this study. Most participants were female (89.45%), with an average age of 20.14 ± 0.69 years. There was no statistically significant difference in the general characteristics between the experimental and control groups (Table 1).

#### 3.1.2. Effect of the Hybrid of Standardized Patient Simulation and Mobile Application Training

Using linear mixed-effects model with the Greenhouse Geisser correction, the results showed statistically significant interactions between group and time for clinical surgical nursing competence (F = 8.68, *p* < 0.001), self-efficacy for clinical performance (F = 13.56, *p* < 0.001), and cultural competence (F = 10.35, *p* < 0.001) (Table 2). Additionally, the results showed that the average clinical surgical nursing competence score was 3.42 (SE = 0.98, 95% CI: 1.48, 5.35, *p* < 0.01) points higher after participants received a hybrid course compared to those who did not receive such a course. The experience participants scored 7.59 points higher at baseline relative to control group (95% confidence interval: 2.75, 12.43, *p* < 0.01) for self-efficacy in clinical performance score. For cultural competence score, there was a significant score difference between the experimental and control group (SE = 1.32, 95% CI: 1.72, 6.95, *p* < 0.01). Figure 2 indicates the time series of variables between the experimental and control groups.

#### 3.1.3. Participants’ Satisfaction with Training

The satisfaction levels of the students who participated in the training intervention were examined to determine the highest and lowest student satisfaction rates in each domain. The highest student satisfaction rate was related to debriefing and reflection (4.25 ± 0.40), and the lowest student satisfaction rate was related to clinical reasoning (4.20 ± 0.44). The overall mean student satisfaction level after attending a surgical nursing course with simulation was 4.25 (0.40) (Table 3).

### 3.2. Qualitative Results

The four focus group discussions were conducted immediately after the intervention with a total of 24 students from the intervention group. Participants were selected using purposive sampling to ensure a mix of genders, religions, academic performance levels, places of residence, and social activities. Upon completion of the hybrid of standardized patients and mobile applications in the surgical nursing course, participants reported significant psychological and professional development. It was not uncommon for them to experience initial feelings of anxiety and unpreparedness following the theory classes. Throughout the simulation learning process, participants underwent a gradual transformation, becoming increasingly prepared and confident. The integrated nature of the course allowed participants to gradually develop the necessary knowledge and skills to succeed in their clinical practice. This course provided participants with the opportunity to combine theoretical learning with practical experience, resulting in comprehensive skills. After completing the training, participants reported higher levels of preparedness and readiness, enabling them to better handle challenges that may arise during clinical practice. Finally, they could apply their skills with confidence during clinical practice. This desirable change demonstrated the outstanding effectiveness of the course.

The analysis results indicated that the effectiveness of the integration of this training is shown through three categories: *educational achievement, dynamic learning experiences, and satisfaction and suggestion* (Figure 3).

*Educational achievement* means that participants self-evaluate their educational achievements in nursing professionalism, empathy, and self-confidence. The participants acknowledged that they had developed various dimensions of nursing professionalism, including cultural competence and knowledge, attitudes, and behavior. Cultural orientation is an essential aspect of nursing professionalism. Our participants developed cultural competence on multiple levels, including cultural awareness, cultural knowledge, cultural sensitivity, and cultural skills. As a result, they gave overwhelmingly positive feedback, expressing that they had significantly improved their cultural competence through this course, which was vital in delivering quality patient care.


*“When I started admitting patients to the hospital, I determined what ethnicity and country they belonged to, so we should search for and learn about their customs, habits, diet, and their way of eating, their beliefs. This knowledge helps us provide better care and meet their needs more effectively.”*
(A5)

To improve their nursing knowledge, attitudes, and behavior, they emphasized the importance of forming a clinical knowledge system, enhancing positive attitudes toward the profession, and developing professional skills such as problem-solving, decision-making, basic nursing, surgical nursing, the nursing process, and health education skills. They also focused on developing soft skills, including communication, teamwork, trust-building, technology, negotiation, and time management skills.

*“After learning about simulation, I feel that it has helped me become more comfortable with the hospital environment and better prepared for unexpected situations I may face in the hospital. I have also learned how to communicate with patients, make judgments about their conditions, diagnose and examine them, as well as visualize theoretical concepts in practice. Additionally, simulation has helped me to practice technical procedures and improve my precision, accuracy, and professional behavior.”* (B4); *and “I have improved my communication skills and learned how to approach and talk to patients, especially difficult ones. I practice patience, even when they are irritated and do not want to talk. I approach them slowly with some simple questions that show my care and concern. This approach helps them open up and feel more comfortable with me.”* (C1).

In addition, it promoted empathy among participants according to a qualitative analysis. They displayed empathy at all levels, with many demonstrating awareness of patients’ feelings and understanding of their thoughts. Emotional, cognitive, behavioral, and moral empathy were all expressed. These findings have significant implications for the development of empathetic skills in professionals.


*“I gained a better understanding of how to interact with real patients at the hospital. I learned how to ask questions more effectively and empathetically and to be more attuned to their feelings of pain and discomfort. Instead of bombarding them with many questions, I now focus on asking the right questions at the appropriate time. I am more patient and understanding when patients are upset or angry, even when they get angry, I do not angry back.”*
(A2)

Self-confidence was reported to increase remarkably due to a combination of theoretical knowledge, simulated practice, and clinical experience. The most prominent areas in which they noticed a significant improvement in confidence levels were communication skills, including teamwork and patient interactions, as well as basic nursing procedures, nursing processes, physical examinations, and other professional activities. Their feedback highlighted the effectiveness of the course in reinforcing their theoretical knowledge, preparing them for clinical practice, and enhancing their self-assurance in the field of nursing.


*“I feel more confident when communicating with patients now. I feel 80–90% more confident than before… Before, I would rarely review the techniques, and when I would go to the clinic, I was afraid of not being able to inject properly. I was also afraid of the clinical instructor yelling at me if I did it wrong, and most importantly, I was afraid of hurting the patients. However, after studying the simulation techniques and reviewing these skills, I feel more confident now.”*
(C8)

*Dynamic learning experience* illustrates that training had a great impact on creating a safe and comfortable learning environment, promoting collaboration, and providing effective training methods that enhance long-term memory and motivation. Many participants felt less stressed and more relaxed without instructors constantly supervising them. The use of models and standard patients for practicing care procedures before treating actual patients not only helped prevent any direct harm but also reduced the pressure on them.


*“In case of any mistakes or gaps in my understanding of the techniques, my teachers and peers are always there to support me and explain the correct way in detail before I proceed to the clinic. In contrast, if I had to immediately start clinical practice right after my theory class, I may make errors while caring for real patients, which could result in punishment.”*
(B6)

It also offered the opportunity to hone their skills by working in groups, collaborating with peers and other healthcare students, supporting one another to solve problems, or handling diverse situations together. This approach capitalized on the benefits of peer learning while allowing instructors to provide guidance and address any queries that might arise throughout the process. At the same time, with this method, groups could observe their peer-solving situations and give each other comments. Taking on different roles helped participants develop many learning strategies and practice many skills.

*“I have become more confident, improved my relationships with friends, and enhanced my skills as a team leader. Overall, I have seen significant progress, and I am excited to continue improving”.* (B2); *and “When I saw my peers perform the same task, I could recognize their mistakes and learn from their experiences. Learning from experience once can help me remember it forever (hahaha). It helps me study better.”* (A1).

*Satisfaction and suggestions:* The participants’ recognition of the significant impact that simulations had on the learning process highlighted the effectiveness of experiential learning methods in education. The course supported improving learner satisfaction in various aspects, such as tutors, teaching methods, scenarios, standardized patients, and facilities. Therefore, they contended that simulations enhance the educational experience by providing an additional layer of understanding through practical application.

*“I want to learn more simulation sessions like this because I feel like I have studied too little, I want to learn more because I found it effective”*. (B6); *“The standardized patient acted very much like a real patient. There were ethnic cases where she spoke exactly like an ethnic minority, asking questions that many ethnic minorities wondered about so we had to think to answer... Our standardized patient expressed all of these concerns well so that we have a chance to practice handling all these situations.”* (A8); *“I know how to make medical records, know nursing assessments and nursing diagnoses better than when I initially learned in class.”* (A1); *“I thought the simulation acts as a transfer station between theory and clinical practice, connecting theory and clinical practice.”* (B1); “*I must admit that the teachers here are incredibly dedicated… The teachers were always available to answer any questions I had, and their willingness to help was greatly appreciated.”* (A1).

In addition, some suggestions were highlighted to improve the quality of the course. These included increasing the simulation duration, providing simulation courses separately, arranging the time between simulation and clinical practice appropriately, and adding more simulation scenarios.

*“It would be better to learn simulation after learning theory and approximately 1–2 weeks before clinical practice.”* (B1); *“Based on my evaluation, the simulation scenarios are quite varied, and they closely resemble real-life situations that students may face in the clinical setting. However, I believe that we need more specific scenarios and more simulation scenarios to learn. It helps us when encountering those situations, not to be distracted or awkward, and know how to build trust with the patient and be more confident in performing the procedure.”* (B4).

## 4. Discussion

This study evaluated the effect of a hybrid of standardized patients and mobile application training on nursing students based on the NLN/Jeffries Nursing Education Simulation Framework. In the baseline phase, the two groups were considered homogenous in terms of their characteristics, clinical surgical nursing competence, self-efficacy in clinical performance, and cultural competence. After the training, the results demonstrated that the use of this teaching strategy provided invaluable practice experiences that contributed to the development of competent practices and cultural competence. Notably, the findings revealed mixed results regarding the experimental group’s foundational understanding of nursing care, as evidenced by the assessment data. The current literature supports the use of simulation as an effective means of enhancing the clinical competence of nursing students. The findings of this study are consistent with this body of research and emphasize the importance of simulation as a critical component of nursing education.

The findings of the study suggested that the hybrid simulation intervention significantly enhanced the clinical surgical nursing competence, cultural competence, and self-efficacy in terms of clinical performance among nursing students (*p* < 0.001). This finding aligned with previous research demonstrating the beneficial impact of simulation-based learning [37,38,39]. Notably, the results of the present study were consistent with those of Hung et al., who similarly reported that simulation-based learning effectively enhances nursing students’ perceived competence, self-efficacy, and satisfaction with learning, particularly following the completion of the ‘Integrated Care in Emergency and Critical Care’ course [40]. To enhance the clinical competence of nursing students, educators, and institutions have undertaken various endeavors to refine simulation-based teaching strategies. These include designing interventions, integrating simulations with other methods, determining session duration and frequency, and incorporating feedback mechanisms. This research provides valuable insights for refining simulation-based teaching strategies in nursing education programs.

According to the qualitative results, the participants provided feedback indicating that the simulation sections were conducive to enhancing the educational impact, including “*educational achievement*”, and “*dynamic learning experiences*”. Jang and Park mentioned that the effect of nursing simulation scenarios promoted an increase in theoretical knowledge, clinical performance skills, and self-confidence [41]. Similarly, previous studies also highlighted that the simulation educational program promoted students’ self-confidence and foundational understanding of how to carry out the nursing process [42]. Jiménez-Rodríguezwe et al. indicated that using standardized patient simulation increases empathy levels as well as humanization competency [11]. Notably, in our results, participants also highlighted increasing technological skills through utilizing mobile healthcare applications. It helps them gain in using technology to develop essential technological competencies and support clinical practice, which is an important part of modern healthcare practice.

The results of our study revealed that the participants expressed a high level of satisfaction with the hybrid simulation training. Additionally, the emphasis on satisfaction and suggestions in the content analysis indicated that the training program was well received and that participants valued the opportunity to provide feedback for continuous improvement. These findings were consistent with earlier studies [43,44], which provided further evidence supporting the effectiveness of hybrid simulation as an effective educational and training tool. Further research is needed to explore the optimal use of simulation as a teaching strategy and to identify the factors that contribute to its success in promoting the development of nursing students’ clinical competence.

The findings of this study might pose some generalization challenges due to several limitations. The participants was recruited from a single institution. Randomization in selecting participant is most of limitation of study. This factor restrict the applicability of our conclusions to other populations and environments.

## 5. Conclusions

This study demonstrated the effectiveness of simulation-based education in preparing nursing students for professional practice. By utilizing a hybrid of standardized patient simulation and mobile healthcare applications, the students gained valuable experience that enhanced their competence. They expressed a high degree of satisfaction with the simulation, as it provided a deeper understanding of theoretical concepts and facilitated the development of clinical skills. While the results are promising, further research is necessary to explore whether students can transfer their knowledge to clinical practice over the long term. Therefore, it is essential to explore the transferability of the knowledge gained by students through rigorous investigation and evaluation.

## Figures and Tables

**Figure 1 ijerph-21-00515-f001:**
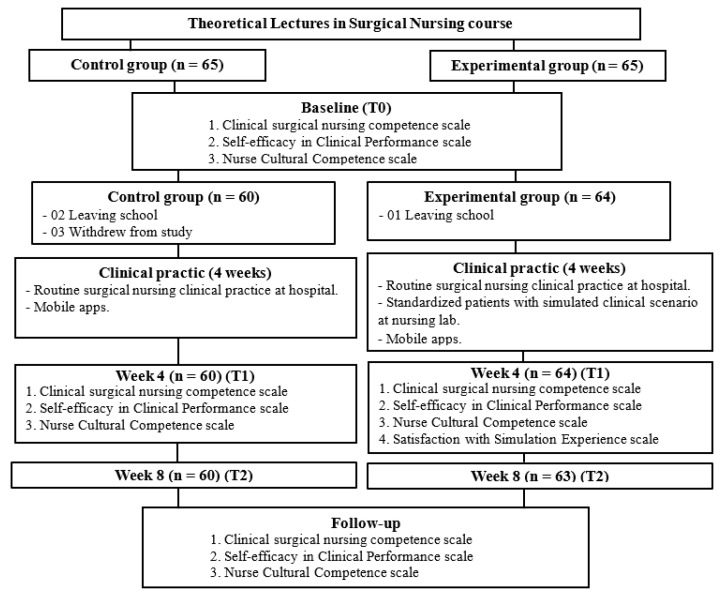
Participant flow through the trial.

**Figure 2 ijerph-21-00515-f002:**
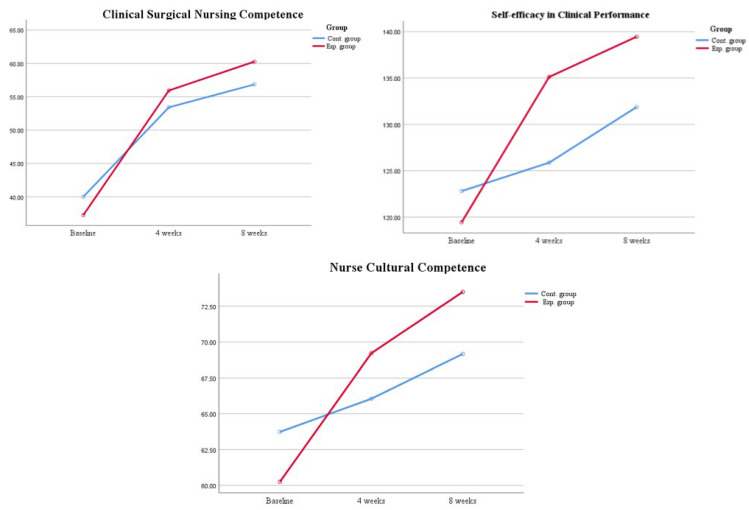
Time series of variables between the experimental and control group.

**Figure 3 ijerph-21-00515-f003:**
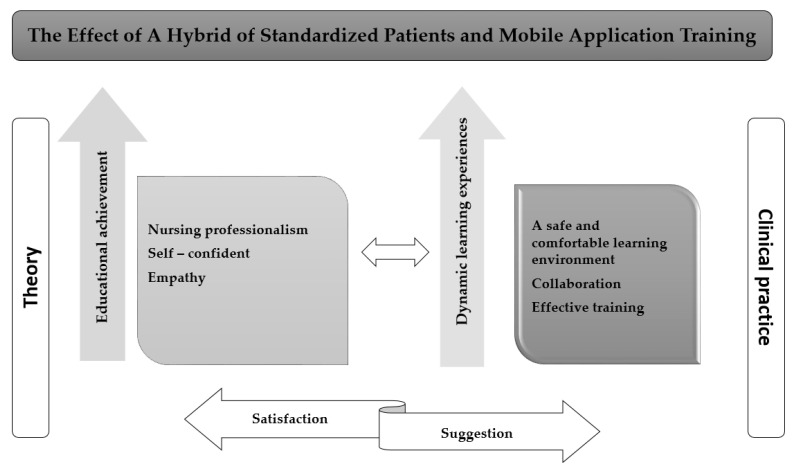
Summaries of qualitative results.

**Table 1 ijerph-21-00515-t001:** General characteristics and homogeneity test between the experimental and control groups (n = 124).

Variables		Cont. (n = 60) n (%)	Exp. (n = 64) n (%)	Test	*p*
Age (years)		20.23 ± 1.18	20.05 ± 0.21	(t) 1.20	0.22 ^a^
Sex	Male Female	7 (11.7) 53 (88.3)	6 (9.4) 58 (90.6)	(χ^2^) 0,17	0.77 ^c^
Ethnicity	Kinh Other	54 (90) 6 (10)	62 (96.9) 2 (3.1)	-	0.15 ^c^
Religion	Buddhism Christianity No religion Other	7 (11.7) 4 (6.6) 49 (81.7) 0	14 (21.9) 3 (4.7) 46 (71.9) 1 (1.5)	-	0.33 ^b^
Taking care of people from other ethnicities and religions	Yes	14 (23.3)	7 (10.9)	-	0.43 ^c^
Living with people from different ethnicities or religions	Yes	14 (23.3)	7 (10.9)	-	0.93 ^c^
Contacting with people from other ethnicities and religions	Yes	13 (21.7)	16 (25)	-	0.68 ^c^
Clinical surgical nursing competence		40.07 ± 9.71	37.25 ± 6.95	(t) 1.87	0.051 ^a^
Self-efficacy in clinical performance		122.80 ± 14.21	119.44 ± 13.41	(t) 1.35	0.67 ^a^
Cultural competence		63.73 ± 8.66	60.25 ± 10.44	(t) 2.03	0.06 ^a^

Exp = experimental group; Cont = control group, ^a^ = test statistic W of the Wilcoxon test, ^b^ = χ^2^-value; ^c^ = *p* value for Fisher’s exact test.

**Table 2 ijerph-21-00515-t002:** The comparison of three scores among two groups (n = 124).

Variable (n = 124)	Group	T0	T1	T2	Source	F	*p*
		Mean ± SD					
Clinical surgical nursing competence	Cont.	40.00 ± 9.58	53.42 ± 7.38	56.85 ± 5.51	Group Time Time × Group	1.27 344.00 8.68	0.26 <0.001 <0.001
	Exp.	37.25 ± 6.95	55.94 ± 6.46	60.27 ± 5.45			
Self-efficacy in clinical performance	Cont.	122.80 ± 14.21	125.87 ± 14.85	131.87 ± 11.91	Group Time Time × Group	4.52 60.88 13.56	0.04 <0.001 <0.001
	Exp.	119.44 ± 13.41	135.11 ± 15.61	139.45 ± 15.21			
Cultural competence	Cont.	63.73 ± 8.66	66.05 ± 7.83	69.16 ± 7.71	Group Time Time × Group	1.35 55.66 10.35	0.25 <0.001 <0.001
	Exp.	60.25 ± 10.43	69.22 ± 7.31	73.50 ± 7.14			

**Table 3 ijerph-21-00515-t003:** Participants’ satisfaction with the intervention.

Characteristics	Mean ± SD	Min–Max
Satisfaction with Simulation Experience	4.25 ± 0.40	3.49–5.00
Debrief and reflection	4.28 ± 0.41	3.44–5.00
Clinical reasoning	4.20 ± 0.44	3.00–5.00
Clinical learning	4.25 ± 0.40	2.50–5.00

## Data Availability

The data that support the findings of this study are available from the corresponding author upon reasonable request.

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
