# Peer review of "Effects of Standardized Patient Simulation and Mobile Applications on Nursing Students’ Clinical Competence, Self-Efficacy, and Cultural Competence: A Quasi-Experimental Study"

_ijerph, 2024, doi:10.3390/ijerph21040515_

Round 1

Reviewer 1 Report

Comments and Suggestions for Authors

Abstract

- please provide the direction of the findings in the abstract. The findings presented are not clear - you are reporting the interactions. As suggested = a table showing the actual scores in the text would improve an understanding of the findings and the way the scores changed over time within and between groups should be summarised in abstract

Methodology

- Not clear when Focus Groups were (P3 L98)

- please indicate what the clinical surgical scale was based on? Eg. existing assessment sheets or literature

- L146 provide more information on what mobile apps were used for and reference this paragraph. In addition on L198 states that mobile appliciton was used for patient data - was this during the simulation - so not real patient data or how was this used?

Data presentation

I woud recommend the following to improve the presentation

Table 1

1)  - Remove Yes and No and only report on the Yes data

2) Add T and X2 before each Test value and change heading to Test

- Provide full text for ADDIE model

- Can you indicate whether your data over time were linked? Did you do paired analysis of changes over time?

3) I would suggest a seperate table to look at the key measures and I would include the full table with items as appendix

4) From the Table non or the findings were significant  but were approaching significance and experimental groups had higher scores

- Can you also present the within group values eg. pre measurement post measurement and last measurement - this is done in graphs but a table with this data woud be useful. 

Table 2

The table presents significant interactions - could be summarised in text and removed 

The figures are most useful but the scale max should be indicated and as suggested actual tables with the data would improve the presentation

Qualitative

Participants information should be provided - how many focus groups? when conducted? who were the participants?

Organisation of the themes/categories can be improved by using each theme as a heading and by indenting the quotes 

More quotes need to be provided to support the categories

General convention

When reporting p-values, it is common to omit the leading zero before the decimal point as p-values cannot exceed 1.0. In addition, please include 3 decimals. In addition p-values are not reported as 0.00 it should be <.001 as 0.00 is not clear

Author Response

Thanks for taking the time to share your views

And please see the attachment.

Regards

Reviewer 2 Report

Comments and Suggestions for Authors

Thanks to the authors for sharing their manuscript. My comments will be unprincipled, but I hope they will be useful:

1.       In section 2.2. “Instruments for training evaluation”, the authors indicate that they developed a clinical surgical nursing competence scale. I would life a more detailed description of this instrument: how was it developed, which labels of the Likert scale were used, did the authors evaluate anything other than internal consistency? In addition, it seems useful to me to give two or three examples of items.

2.       It seems to me that a table can also be used to describe qualitative results and systematize the responses of respondents.

3.       As limitations of the study, the authors highlight the small sample size of students and their affiliation to a single university. It seems to me that it is also worth pointing out here that the limitation of the study is a quasi-experimental design, and the prospect of the study is a randomized controlled trial.

My comments do not detract from the importance of the manuscript, I consider it suitable for publication.

Author Response

Your comments mean a lot to us. Thanks for taking the time to share your views.

Best wishes
